# MDT-A2G: Exploring Masked Diffusion Transformers for Co-Speech Gesture Generation

Xiaofeng Mao*
xfmao23@m.fudan.edu.cn
Fudan University
Shanghai, China

Zhengkai Jiang*
zkjiang@ust.hk
Tencent Youtu Lab
Shanghai, China

Qilin Wang
Chencan Fu
qlwang22@m.fudan.edu.cn
chencan.fu@zju.edu.cn
Fudan University
Zhejiang University
China

Jiangning Zhang
Jiafu Wu
vtzhang@tencent.com
jiafwu@tencent.com
Tencent Youtu Lab
Shanghai, China

Yabiao Wang
caseywang@tencent.com
Zhejiang University
Tencent Youtu Lab
China

Chengjie Wang
jasoncjwang@tencent.com
Tencent Youtu Lab
Shanghai, China

Wei Li
liwei.yxgh@vivo.com
Vivo Communication
Technology Co. Ltd
Shanghai, China

Mingmin Chi†
mmchi@fudan.edu.cn
Fudan University
Shanghai, China

## Abstract

Recent advancements in the field of Diffusion Transformers have substantially improved the generation of high-quality 2D images, 3D videos, and 3D shapes. However, the effectiveness of the Transformer architecture in the domain of co-speech gesture generation remains relatively unexplored, as prior methodologies have predominantly employed the Convolutional Neural Network (CNNs) or simple a few transformer layers. In an attempt to bridge this research gap, we introduce a novel Masked Diffusion Transformer for co-speech gesture generation, referred to as MDT-A2G, which directly implements the denoising process on gesture sequences. To enhance the contextual reasoning capability of temporally aligned speech-driven gestures, we incorporate a novel Masked Diffusion Transformer. This model employs a mask modeling scheme specifically designed to strengthen temporal relation learning among sequence gestures, thereby expediting the learning process and leading to coherent and realistic motions. Apart from audio, Our MDT-A2G model also integrates multi-modal information, encompassing text, emotion, and identity. Furthermore, we propose an efficient inference strategy that diminishes the denoising computation by leveraging previously calculated results, thereby achieving a speedup with negligible performance degradation. Experimental results demonstrate that MDT-A2G excels in gesture generation, boasting a learning speed that is over 6× faster than traditional

diffusion transformers and an inference speed that is 5.7× than the standard diffusion model. Our code is available at MDT-A2G.

## CCS Concepts

• **Human-centered computing** → **Human computer interaction (HCI)**; • **Computing methodologies** → *Motion processing*.

## Keywords

Gesture Generation, Motion Processing, Data-Driven Animation, Masked Diffusion Transformer

**ACM Reference Format:**
Xiaofeng Mao, Zhengkai Jiang, Qilin Wang, Chencan Fu, Jiangning Zhang, Jiafu Wu, Yabiao Wang, Chengjie Wang, Wei Li, and Mingmin Chi. 2024. MDT-A2G: Exploring Masked Diffusion Transformers for Co-Speech Gesture Generation. In *Proceedings of the 32nd ACM International Conference on Multimedia (MM '24), October 28–November 1, 2024, Melbourne, VIC, Australia.* ACM, New York, NY, USA, 9 pages. https://doi.org/10.1145/3664647.3680684

## 1 Introduction

The objective of co-speech gesture generation is to synthesize human gestures from audio and other modalities. Its significance is escalating in the development of virtual avatars and interactive technologies. In everyday life, there is a substantial demand for the efficient production of high-quality and diverse human gesture animations. Consequently, the production of high-quality and diverse gestures at an affordable cost plays a pivotal role in practical applications.

With the rapid development of generative models such as Variational Autoencoders (VAEs), Generative Adversarial Networks (GANs), and the more recent Diffusion Models (DMs), numerous works [2, 3, 6, 14, 16, 21–24, 31, 34–38, 40, 41] have proposed utilizing these powerful foundational models for the task of co-speech gesture generation. Although these works have achieved good generative quality, we find the following limitations in practical applications: **1) Poor diversity.** For VAE-based methods [15, 19, 38], the goal is to learn a meaningful latent space where similar inputs are mapped close together. However, if the model relies too much on the

---

*Both authors contributed equally to this research.

†Corresponding author (This work was supported by Natural Science Foundation of China under contract 62171139).

---

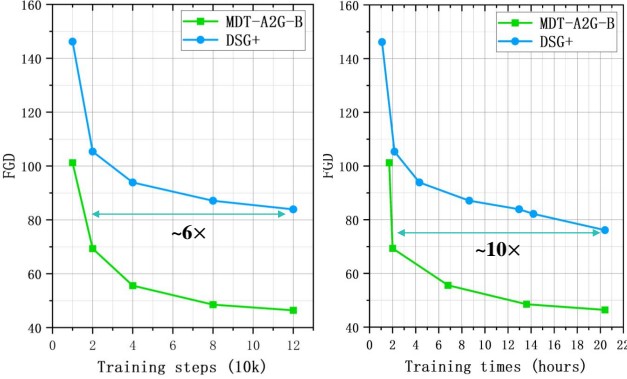

**Figure 1: Comparison between DSG+ [37] and our MDT-A2G-B with respect to training steps/times on a single A100 GPU. Compared to DSG+, MDT-A2G-B exhibits a faster training convergence speed and superior performance, demonstrating the effectiveness of proposed method.**

decoder's capacity to reconstruct the data from the latent variables, it might ignore the latent variables, leading to a phenomenon known as posterior collapse. This results in a lack of diversity in the generated samples. For GAN-based methods [11, 23], training instability and mode collapse issues are common, leading to the generator producing a limited variety of human gestures, or even identical ones. **2) Inefficiency.** DM-based methods [2, 3, 6, 16, 31, 34, 36, 37, 40] require multiple iterations during the generation process. Each iteration involves a forward pass through the model, resulting in typically long inference time. Besides, current DMs applied in A2G domain often struggle to effectively learn the semantic relationships between features of different modalities, leading to slow training convergence.

To address above challenges, we propose **MDT-A2G**, a **M**asked **D**iffusion **T**ransformer framework designed for **A**udio-**t**o-**G**esture task. Sepecifically, we leverage the contextual reasoning capability of mask modeling and combine it with DMs to ensure the quality and diversity of co-speech gesture synthesis. During training, the masked modeling denoising network takes multi-modal conditions and partially masked noisy features as input and outputs original gestures. As shown in Figure 1, benefited by the masked modeling's strong semantic learning ability, our MDT-A2G can generate human gestures that are both realistic and diverse. The convergence of the learning process is also accelerated. To the best of our knowledge, this is the first attempt of mask modeling DMs in A2G task. We also propose the incorporation of a simple-yet-effective multi-modal feature fusion module into existing architecture. This module is carefully designed for fusing various modalities, simplifying the complex feature processing and fusion mechanism. To improve the efficiency of inference, we introduce an efficient inference strategy during sampling without compromising the quality and diversity of the generated gestures.

Extensive qualitative and quantitative experiments demonstrate that our proposed MDT-A2G can produce **high-quality**, **diverse** and **temporally-coherent** human gestures **in a high efficiency**. We achieve state-of-the-art performance regardless of upper and full body, outperforming existing co-speech generation methods. The main contributions can be summarized as follows:

- We propose MDT-A2G, a masked diffusion transformer framework designed for audio-to-gesture task, enhancing semantic understanding of diverse modalities and hastening training convergence that is 6 × faster.
- We introduce a novel sampling acceleration technique that significantly reduces the inference time, achieving a speedup of 5.7× faster compared to the standard diffusion model.
- Extensive experiments demonstrate that our proposed approach achieves state-of-the-art performance both qualitatively and quantitatively.

## 2 Related Work

### 2.1 Co-speech Gesture Generation

The field of co-speech gesture generation, which concentrates on the production of gestures that synchronize with speech audio, has witnessed the advent of numerous learning-based methodologies. Among these, the hierarchical approach proposed by Liu et al. [24] is significant, as it considers both speech semantics and the structure of human gestures. Yoon et al. [39] have also made substantial contributions by utilizing a recurrent neural network and multi-modal context to devise a translation-based approach. Liu et al. further enriched the field by introducing the BEAT dataset and the Cascaded Motion Network (CaMN) [23]. Other notable contributions include EMAGE [22], a method that integrates Masked Gesture Reconstruction and Audio-Conditioned Gesture Generation, and TalkSHOW [38] by Yi et al., which employs an autoencoder and a VQ-VAE for the generation of face and body motions. In recent years, diffusion models, celebrated for their capacity to model intricate data distributions and perform many-to-many mappings, have been increasingly harnessed for gesture synthesis. A number of studies [1, 14, 31, 35, 36, 40, 41] have focused on the generation of co-speech gestures using diffusion models, introducing various strategies to augment motion generation and style control. Despite these advancements, existing works continue to struggle with the challenge of harmonizing diversity, authenticity, and speed in the generation of gestures from speech.

### 2.2 Diffusion Transformers

Diffusion Transformers have made significant advancements in the generation of high-quality images [4, 29], videos [25], and 3D models [27]. DiT [29] is among the pioneering works to explore transformer-based architectures in the field of image generation. It introduces innovative designs to train latent diffusion models by replacing the commonly used U-Net with a transformer backbone. PixArt-$\alpha$[4] proposes the incorporation of cross-attention modules into the Diffusion Transformer (DiT) to inject text conditions, thereby achieving photorealistic text-to-image synthesis. DiT-3D [27] explores the use of plain diffusion transformers for 3D generation, which directly operate the denoising process on voxelized point clouds instead of latent spaces. The Masked Diffusion Transformer (MDT) [9] introduces a masked latent modeling scheme, enhancing the ability to learn contextual relations among semantic parts. Like the Masked Autoencoder (MAE) [12], MDT also adopts an asymmetric transformer to predict masked tokens

from unmasked ones while simultaneously undergoing the diffusion generation process. As a result, it achieves superior image synthesis performance with a state-of-the-art (SOTA) Frechet Inception Distance (FID) score on the ImageNet dataset and faster learning speed. In this paper, we aim to explore such a masked scheme in the domain of co-speech gesture generation. Our MDT-A2G directly operates the denoising process on the gesture and incorporates multi-modal information, including speech, emotion, and identity, to align with the gesture.

## 2.3 Mask Modeling

The masked scheme first demonstrated its effectiveness in Natural Language Processing (NLP), with BERT-based models [8, 17] enhancing the performance of word embeddings through masked language modeling and transformer architecture. Subsequently, the Masked Autoencoder (MAE) [12] extended the masking strategy into the field of computer vision and developed an asymmetric encoder-decoder architecture with a high mask ratio. This concept of masked representation learning has been employed in other modalities as well, such as video [32], audio [13], and time series [7]. Most related to our work, EMAGE [22] proposes a temporal transformer to learn robust motion representation for gesture generation. In contrast, we resort to masking strategy to strengthen temporal relation learning sequence gesture, resulting faster training speed and more realistic gesture generation.

## 3 Preliminaries

### 3.1 Gesture Format

We employ the BEAT dataset [23], recognized for being the most comprehensive motion capture dataset in terms of duration and the diversity of modalities it covers. The BEAT dataset stores motion capture data in the BVH file format, articulating motion representation through Euler angles. Following DSG+ [37], we favor using rotation matrices over Euler angles to represent joint rotations. We leverage 75 joints including 27 body joints, 48 hand joints for whole-body gesture generation and choose a subset of 14 upper-body joints, along with the 48 hand joints, for generating upper-body gestures.

### 3.2 Diffusion Model

We consider starting the denoising process with pure noise and generating gestures using a diffusion model. We introduce then denoising diffusion probabilistic models (DDPM). DDPM defines a $T$-step forward process and a $T$-step reverse process. DDPM turns the present state $x_0$ into the previous state $x_t$ by gradually adding random noise through the forward process. The diffusion process formulas is as follows:

$$q(x_t|x_0) = \mathcal{N}(x_t, \sqrt{\overline{\alpha}_t}x_0, (1 - \overline{\alpha}_t)\mathbf{I}),$$
$$x_t = \sqrt{\overline{\alpha}_t}x_0 + \sqrt{1 - \overline{\alpha}_t}\epsilon, \epsilon \sim \mathcal{N}(0, \mathbf{I}). \quad (1)$$

where $x_t$ is the noised image at time-step $t$, $\overline{\alpha}_t$ is the predefined scale factor, and $\mathcal{N}$ represents the Gaussian distribution.

It means that the original state $x_0$ is transformed into $x_t$ by gradually adding Gaussian noise.

Conversely, the reverse denosing phase is designed to revert the noise-infused data back to its original structured form, effectively reconstructing the joint distribution $p_\theta(x_{0:T})$. This phase is defined by:

$$p_\theta(\boldsymbol{x}_{0:T}) = p_\theta(\boldsymbol{x}_T) \prod_{t=1}^{T} p_\theta(\boldsymbol{x}_{t-1}|\boldsymbol{x}_t),$$
$$p_\theta(\boldsymbol{x}_{t-1}|\boldsymbol{x}_t) = \mathcal{N}(\boldsymbol{x}_{t-1}; \mu_\theta(\boldsymbol{x}_t, t), \Sigma_\theta(\boldsymbol{x}_t, t)). \quad (2)$$

Here, the model assumes a constant time-dependent variance $\Sigma_\theta(\boldsymbol{x}_t, t) = \beta_t I$. A generative model $\mathcal{G}_\theta$ is then formulated to estimate the mean of the Gaussian distribution. For scenarios requiring conditional generation, the conditional variable **c** is seamlessly incorporated into the model's architecture.

We follow [37,44,49] to predict the signal itself instead of predicting $\epsilon_\theta(x_t, t)$ [18]. The network $D$ learns parameters $\theta$ based on the input noise $x_t$, noising step $t$ and conditions $c$ to reconstruct the original signal $x_0$ as :

$$\hat{x}_0 = D(x_t, t, c) \quad (3)$$

## 4 Methodology

### 4.1 Problem Formulation

The task of co-speech gesture generation can be formulated as a sequence-to-sequence learning problem. Given an input sequence of speech features and other modal conditions, the goal is to generate a corresponding sequence of human gestures. The model needs to consider the temporal alignment between the speech and gesture sequences, as well as the incorporation of multi-modal information (e.g., emotion, speaker ID and text transcript) to generate more realistic and diverse gestures.

### 4.2 Overall Framework

We propose a novel framework named MDT-A2G for generating human gestures from multi-modal conditions. The overall training and sampling pipeline is illustrated in Figure 2. Specifically, gestures are generated based on the noisy gesture $x_t$, audio $x_a$, text $x_{txt}$, ID $x_s$, emotion $x_e$, and time steps $t$. We first obtain the input noisy gesture $x_t$ and specific multi-modal condition. Then, we fuse the multi-modal feature to get $x_{fuse}$ as follows:

$$\hat{t} = MLP(E_t(t))),$$
$$\hat{x}_t = \text{Concat}[x_t, E_s(x_s) + \hat{t}, E_e(x_e) + \hat{t}, E_{txt}(x_{txt}), E_a(x_a)], \quad (4)$$
$$x_{fuse} = Attention(\hat{x}_t).$$

where $E(\bullet)$ represents the embedding layer. $Attention(\bullet)$ utilizes cross-local attention used in DSG+ [37]. We mask a portion of $x_{fuse}$ to obtain $x_u$, which is then fed into the transformer encoder to map it to a latent representation. We then concatenate the latent representation with multiple sets of learnable tokens $x_l$ and then feed them into a self-attention (SA) block for feature integration.

$$\hat{x}_u = (1 - MASK) * x_u + MASK * SA(\text{Concat}[x_l, Encoder(x_u)]) \quad (5)$$

Finally, we pass $\hat{x}_u$ through the transformer decoder to obtain the final reconstructed features:

$$\hat{x}_0 = Decoder(\hat{x}_u) \quad (6)$$

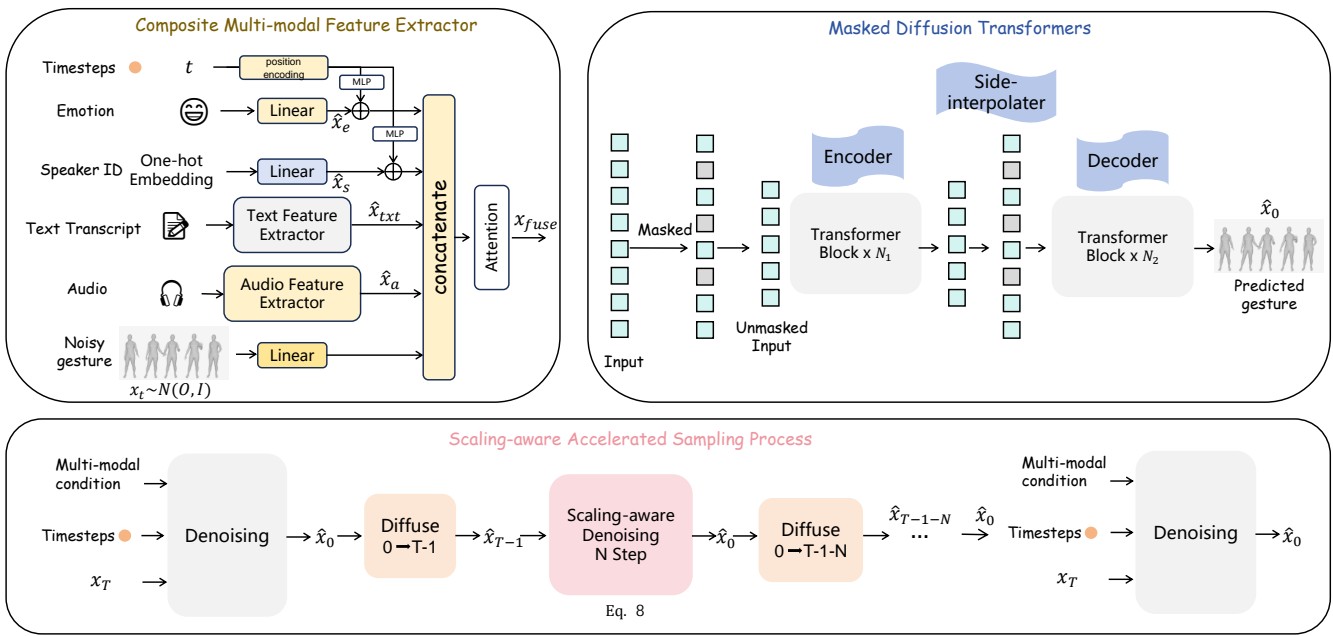

**Figure 2: Overview of MDT-A2G. It primarily consists of three components: (1) Composite Multi-modal Feature Extractor, (2) Masked Diffusion Transformers, and (3) Scaling-aware Accelerated Sampling Process. For the multi-modal feature extractor, we propose an innovative feature fusion strategy that integrates time embeddings with emotion and ID features. These will be further concatenated with text, audio, and gesture features, resulting in a comprehensive feature representation. Additionally, we have designed a Masked Diffusion Transformer structure to expedite the convergence of the denoising network, thereby leading to more coherent motions. Finally, we introduce a scaling-aware accelerated sampling process by utilizing diffused results from previous timesteps, resulting in a faster sampling process.**

### 4.3 Composite Multi-modal Feature Extraction

Previous methods, such as DSG+, handle features from different modalities in a complex manner, repeatedly concatenating information from different conditions. We believe that this approach hinders the learning of features from different modalities. Therefore, we have simplified the feature fusion process compared to former methods. Our feature fusion strategy is simple-yet-effective, requiring only a single concatenation process.

**Obtaining specific conditions.** The generation of human gestures is dependent on a set of specific conditions, which are extracted as follows. For audio $x_a$, the raw input is resampled at 16 kHz, and a suite of features including Mel Frequency Cepstral Coefficients (MFCCs), Mel spectrogram features, rhythmic features, and onset points are computed. These features are then integrated with features extracted by the pre-trained WavLM [5] model to form the final audio feature representation $\hat{x}_a$. For text $x_{txt}$, word embeddings are obtained using the pre-trained FastText [26] model. A linear layer is then used to generate the final text feature $\hat{x}_{txt}$. For ID $x_s$, the ID information is encoded as a one-hot vector and transformed into an ID embedding $\hat{x}_s$ via a linear layer. Similarly, the emotional state $x_e$ is encoded as a one-hot vector and transformed into an emotion feature $\hat{x}_e$. During training, the noise step $t$ is randomly selected from a uniform distribution ranging from 1 to $T$. After position encoding, $t$ is processed by a MLP to yield the time feature $\hat{t}$. The noisy gesture $x_t$ is sampled from a standard

normal distribution $\mathcal{N}(0, \mathbf{I})$ during training and sampling stages. Then, $x_t$ is dimensionally reduced through a linear layer. During the training process, $x_e$ and $x_s$ is randomly masked according to a Bernoulli distribution. The temporal embedding $x_t$ is then added to the ID and emotion features after passing through an MLP layer, resulting in the fused feature.

**Multi-modal Feature Fusion.** The process of feature fusion involves the concatenation of the aforementioned conditions and noisy gestures along the feature dimension. This concatenated feature is then subjected to a cross-local attention module [37], which serves to fuse the different modalities effectively, allowing the model to understand the complex relationships between speech, text, style, emotion, and gestures. This is particularly important in co-speech gesture generation, as gestures are often closely related to the content and emotion of the speech. Furthermore, the inclusion of emotion as an additional condition allows the model to generate gestures that are not only contextually appropriate but also emotionally congruent with the speech.

### 4.4 Masked Diffusion Transformers

Previous diffusion model methods on A2G tasks do not pay attention to the importance of masked modeling, which result in slow learning of the semantic correlations among different modal features and slow training convergence. Inspired by the success of masking scheme on video [32], image [12], we utilize this strategy

on gesture generation task, leading to more coherent motion generations. We mask the input features and then feed them into multiple Transformer Blocks, which are divided into encoders, decoders, and side-interpolators. We adopt an asymmetric design that allows the encoder to only take in unmasked features, reducing computational cost.

**Masking Operation.** Given a feature matrix $x_{fuse} \in \mathbb{R}^{N \times d}$, where $d$ signifies the number of channels and $N$ denotes the number of tokens, we randomly conceal tokens at a rate $\rho$. This results in a set of unmasked tokens, denoted as $x_u \in \mathbb{R}^{d \times \hat{N}}$, where $\hat{N} = (1 - \rho)N$. To track which tokens are masked, we create a binary mask $MASK \in \mathbb{R}^N$ with ones assigned to the masked positions. The unmasked tokens $x_u$ are subsequently processed. Using only the unmasked tokens $x_u$ helps to reduce computational overhead.

**Encoder.** For the encoder, we only map the visible unmasked features to latent representations, which allows us to use fewer computations. Different masking rates affect the experimental results, and we will discuss them in detail in the experimental section.

**Side-interpolater.** Inspired by [9], the latent representationS is processed through the side-interpolator, which restores it to its original feature dimensions. As illustrated in Figure 2, the side-interpolator, a compact network, leverages the encoder's output to estimate the masked tokens during training, working solely with the unmasked tokens $x_u$. During inference, all tokens $x_u$ are processed, creating a disparity in the number of tokens. To maintain consistency, the side-interpolator fills masked positions with a shared learnable token, generating an interpolated embedding $x_I$, computed as $x_I = SA(\text{Concat}[x_l, Encoder(x_u)])$. A masked short-cut connection combines $x_u$ and the interpolated embedding $x_I$ into $\hat{x}_u = (1 - MASK) \cdot x_u + MASK \cdot x_I$. Within the side-interpolator, the unmasked features are concatenated with multiple sets of learnable latent mask tokens $x_l$, which assist in restoring the original feature size, before being fed into a Transformer block for feature interaction.

**Decoder.** The decoder receives the output from the side-interpolator, which comprises the encoded latent representation along with the latent mask tokens. Ultimately, these side-interpolator outputs are passed to the decoder for reconstruction. These decoders are less deep compared to the encoders. In the training stage, the encoder, decoder, and side-interpolator work in tandem to forecast the masked features based on the unmasked features. During inference, the side-interpolator is no longer utilized.

## 4.5 Training Objective

To train our networks, we employ the Huber loss as the gesture loss $\mathcal{L}g$:

$$\mathcal{L}_g = E_{x_0 \in q(x_0|c), t \sim [1,T]}[\text{HuberLoss}(x_0 - \hat{x}))]. \quad (7)$$

Following the DDPM denoising process, we predict the gesture $\hat{x}_0$ at each time step $t$, as depicted in Figure 2. During the training process, we provide both the complete feature $x_{fuse}$ and the masked tokens $x_u$ to the diffusion model. This approach ensures that the model does not overly focus on reconstructing the masked region while neglecting the diffusion training. The additional costs associated with using the masked latent embedding are minimal.

This is further supported by the findings in Figure 1, which demonstrate that MDT-A2G-B achieves faster learning progress in terms of total training hours compared to the DSG+.

## 4.6 Scaling-aware Accelerated Sampling Process

Previous diffusion models [31, 37] in the Audio-to-Gesture (A2G) domain have not fully explored the possibilities of skip sampling techniques similar to DDIM [30], which can significantly expedite the sampling process. However, its direct application is problematic as it exacerbates the diffusion model's exposure bias [28], the discrepancy between the sampled $\hat{x}_t$ and the training's $x_t$. Assume that $\hat{x}_0^t$ represents the original gesture predicted by the diffusion model at step $t$, to mitigate this issue, we propose the following method:

THEOREM 4.1. *Given $\hat{x}_0^t$ and the sampled $\hat{x}_t$ at time $t$, we can compute $\hat{x}_0^{t-1}$ for time t-1 without relying on neural networks. The subsequent formula is employed to diminish the exposure bias:*

$$\hat{x}_0^{t-1} = (\hat{x}_t - \sqrt{1 - \overline{\alpha}_t} * \frac{(\hat{x}_t - \sqrt{\overline{\alpha}_t}\hat{x}_0^t)/\sqrt{1 - \overline{\alpha}_t}}{scale})/\sqrt{\overline{\alpha}_t}. \quad (8)$$

*Here, scale is a hyperparameter greater than 1 but close to 1.*

The proof is detailed in Appendix A. Our approach incorporates a 1:N accelerated sampling method, where the current step is fully processed and the next N steps are expedited through skip sampling. It is crucial to note that this strategy is distinct from distillation-it maintains the network's performance during full sampling without necessitating extra training, and does not introduce additional computational overhead, thus offering a straightforward and practical solution.

## 4.7 Model structure

We have set different number of blocks for our MDT-A2G and designed variants with varying hyperparameters (e.g., number of layers and heads). We denote these variants of our model as MDT-A2G-XS (extra small), MDT-A2G-S (small), MDT-A2G-B (base), and MDT-A2G-L (large). Among them, the whole parameter size of MDT-A2G-B is comparable to that of DSG+. The specific network configurations are detailed in Appendix E.

## 5 Experiments

### 5.1 Settings

**Datasets.** We utilize the BEAT [23] dataset, which includes 120Hz motion capture data and audios from 30 speakers, featuring 10-minute conversations and 1-minute self-talks. In addition to motion and audio, BEAT includes extra information such as text, identity, emotion annotations, and facial expressions. Following [33], we select 1-hour audio per speaker and split the dataset 70% for training, 10% for validation and 20% for testing.

**Metrics.** We use Frechet Gesture Distance (FGD) [39] to evaluate the quality of the generated gestures, which uses a pretrained gesture feature extractor and calculates the Frechet distance between the distribution of the features of real and generated gestures. Following [23], we use Semantic-Relevant Gesture Recall (SRGR) [23] and Beat Alignment Score (BeatAlign) [20] to evaluate the diversity

**Table 1: Quantitative comparison with SOTAs. →: represents a closer proximity to the ground truth, indicating better performance.**

|  | Method | FGD↓ | Diversity→ | SRGR↑ | BeatAlign→ |
|---|---|---|---|---|---|
| Upper Body | Ground Truth | - | 395.20 | - | 0.894 |
|  | CaMN [23] | 79.24 | 295.84 | **0.216** | 0.828 |
|  | MDM [31] | 94.91 | 305.73 | 0.209 | 0.831 |
|  | DSG+ [37] | 87.91 | **421.74** | 0.212 | 0.842 |
|  | MDT-A2G-B (Ours) | **77.79** | 348.05 | 0.214 | **0.876** |
| Whole Body | Ground Truth | - | 395.20 | - | 0.893 |
|  | CaMN [23] | 57.46 | 304.49 | **0.241** | 0.821 |
|  | MDM [31] | 92.37 | 337.42 | 0.231 | 0.812 |
|  | DSG+ [37] | 83.91 | 432.16 | 0.236 | 0.840 |
|  | MDT-A2G-B (Ours) | **46.42** | **381.33** | 0.240 | **0.871** |

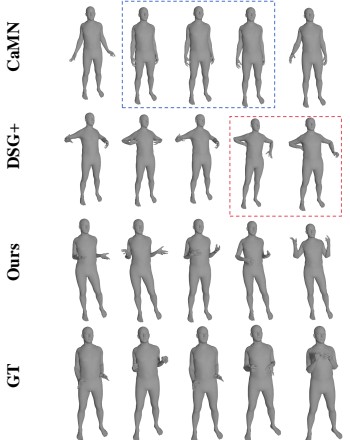

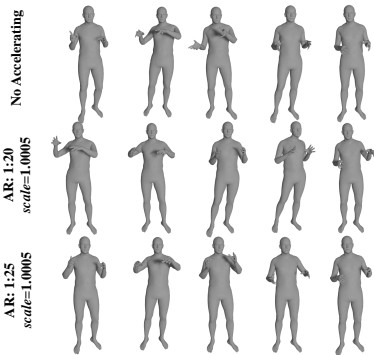

**Figure 4: Comparison with different acceleration ratio. AR is Acceleration ratio. All use MDT-A2G-B**

**Figure 3: Qualitative comparison of whole motion generation. Refer to the supplementary video for a more intuitive comparison.**

and synchrony of the generated gesture. SRGR enhances the traditional Probability of Correct Keypoint (PCK) [19] by introducing semantic relevance scoring as weights. BeatAlign is a measurement between audio and gesture beats through Chamfer Distance [20], which reflects the similarity between audio and gesture beats. We also use Diversity Score [18] to measure the diversity of the generated gestures. The diversity score is measured of the average feature distance [18]. We use the pretrained autoencoder used for FGD.

**Implementation Details.** The motion data is downsampled to 30 Hz from original 120 Hz and we use 300 frames for training. Following [2], we use rotation matrix instead of the original Euler angle for rotation representation. We train our model for 120k steps with a batch size of 150 and use AdamW optimizer at a learning rate of 3e-5. All the experiments are conducted on a single NVIDIA A100 GPU.

**Comparison Methods.** We compare our proposed method with state-of-the-art gesture generation methods, including DSG+ [37], CaMN [23] and MDM [31]. For a fair comparison, we retain all these

methods in the same setting. Specifically, we retrain CaMN to use audio, text, emotion and speaker identity as conditions and DSG+ to use seed gestures, audio and speaker identity as conditional inputs. For MDM, we retrain it to take audio features as input. Addition to whole body gesture generation, we also conduct experiments to compare the results of upper body gesture generation.

## 5.2 Quantitative Comparison

Table 1 presents the quantitative results. Our approach surpasses other methods in generating both upper body and full-body motions. While masked modeling greatly improves the realism of the generated gestures, it does not necessarily enhance the Diversity Score. It's important to note that a higher Diversity Score is not always inherently beneficial. Random noise may achieve a high Diversity Score, but it is not the target we aim for. Additionally, the inherent randomness in the sampling of diffusion models can negatively affect the SRGR metric, which focuses on the accuracy of generated gestures. However, creating realistic motions does not always require strict conformity to ground-truth movements. Nonetheless, our method effectively reduces the adverse impacts on the SRGR metric. Furthermore, we have discovered that employing masked modeling can enhance the Beatalignment score between audio and gestures, thereby improving the Beatalignment

**Table 2: Quantitative comparison with different accelerating configurations.**

| Method | Acceleration ratio | *scale* | Average Time(s)↓ | FGD→ | Diversity→ | SRGR↑ | BeatAlign→ |
|---|---|---|---|---|---|---|---|
| MDT-A2G-B | No Accelerating | 1 | 8.711 ± 1.495 | 46.42 | 381.33 | 0.240 | 0.871 |
| MDT-A2G-B | 1:20 | 1 | 1.984 ± 0.214 | 59.93 | 334.88 | 0.240 | 0.872 |
| MDT-A2G-B | 1:20 | 1.0005 | 1.984 ± 0.215 | 57.61 | 340.11 | 0.240 | 0.872 |
| MDT-A2G-B | 1:25 | 1 | 1.567 ± 0.196 | 64.17 | 326.28 | 0.240 | 0.873 |
| MDT-A2G-B | 1:25 | 1.0005 | 1.587 ± 0.204 | 61.81 | 331.12 | 0.240 | 0.873 |

**Table 3: Ablation study on the proposed components.**

| CMFE | Masked | FGD Score↓ | Diversity→ | SRGR↑ | BeatAlign→ |
|---|---|---|---|---|---|
| × | × | 83.91 | 432.16 | 0.236 | 0.840 |
| ✓ | × | 54.62 | **387.68** | 0.238 | 0.863 |
| × | ✓ | **45.68** | 365.8 | 0.240 | 0.869 |
| ✓ | ✓ | 46.42 | 381.33 | **0.240** | **0.871** |

**Table 4: Ablation study on model scale.**

| Name | FGD↓ | Diversity→ | SRGR↑ | BeatAlign→ |
|---|---|---|---|---|
| MDT-A2G-TS | 95.85 | 335.71 | **0.241** | **0.872** |
| MDT-A2G-S | 64.39 | **392.84** | 0.240 | 0.870 |
| MDT-A2G-B | **46.42** | 381.33 | 0.240 | 0.871 |
| MDT-A2G-L | 50.27 | 390.27 | 0.238 | **0.872** |

**Table 5: Ablation study on different wrider mask ratios.**

| Method | Mask Ratio | FGD↓ | Diversity→ |
|---|---|---|---|
| MDT-A2G-B | 0.1 | 49.54 | 376.77 |
| MDT-A2G-B | 0.2 | 48.38 | **382.46** |
| MDT-A2G-B | 0.3 | 47.74 | 381.14 |
| MDT-A2G-B | 0.4 | **46.42** | 381.33 |
| MDT-A2G-B | 0.5 | 48.47 | 376.88 |
| MDT-A2G-B | 0.6 | 50.46 | 372.06 |
| MDT-A2G-B | 0.7 | 49.38 | 366.84 |
| MDT-A2G-B | 0.8 | 52.28 | 373.49 |

**Table 6: Ablation study on feature process.**

| Structure | Name | FGD↓ | Diversity→ |
|---|---|---|---|
| MDT-A2G-B | baseline | 45.68 | 365.8 |
| MDT-A2G-B | AdaLN-Zero | 82.57 | 319.23 |
| MDT-A2G-B | Ours | **46.42** | **381.33** |

score. This approach compels the denoising network to learn the correlated features between multi-modal conditions and gestures.

## 5.3 Qualitative Results

In the visual representation of our experimental outcomes illustrated in Figure 3, we clearly delineate the distinctions between the competing methodologies and our proposed approach. The CaMN model employs a straightforward methodology that merely concatenates multimodal inputs. While this results in technically correct outputs, the resultant actions are notably monotonous and lack dynamic variation, as highlighted within the blue box of the figure. This method, while reliable, does not allow for the generation of nuanced or complex motion patterns, leading to a visually static presentation. On the other hand, DSG+ leverages an advanced diffusion model coupled with self-attention mechanisms, aimed at enhancing the diversity of generated actions. This model is capable of producing a wide array of actions, thereby enriching the dynamic quality of the outputs. However, this complexity comes with its own set of challenges, as there is a tendency to generate anatomically incorrect gestures or unrealistic motions, as indicated by the examples in the red box. While the ambition of DSG+ to achieve high variability is commendable, it sometimes sacrifices accuracy and realism in its generated actions.

In contrast, our proposed methodology synthesizes the strengths of both the CaMN and DSG+ approaches while mitigating their weaknesses. By integrating a refined feature extraction mechanism with a robust diffusion process, our model not only ensures the

generation of diverse actions but also maintains a high degree of realism and accuracy. The actions generated by our model are both varied and realistic, effectively capturing the subtleties and complexities of human motion without the generation of abnormal gestures. More detailed qualitative comparison is shown in the appendix.

## 5.4 Ablation Studies

**Effectiveness of diffrent model scale.** The impact of different scale models on the outcomes has been extensively validated in the field of image generation. However, previous methods have not explored the impact of model scalability on the audio-to-gesture task. As we utilize Masked Diffusion Transformer as backbone, which possesses excellent scalability properties, we compare the effects of different network sizes on gesture generation here. Table 4 shows the quantitative results.

**Effectiveness of sampling acceleration.** Table 2 illustrates the impact of various acceleration ratios. Notably, at an acceleration ratio of 1:20 and 1:25. We do not observe significant increases in FGD Score and decreases in Diversity Score. Meanwhile, the SRGR remain relatively stable. Furthermore, we find that introducing the Scaling-aware Accelerated Sampling Process (*scale* > 1) yield better results. Qualitative comparison is shown in Figure 4.

**Table 7: Ablation study with MDT-A2G-B.**

| (a) Effect of Decoder Depth. | |
| --- | --- |
| Decoder Depth | FGD↓ |
| 2 | **46.42** |
| 4 | 46.69 |

| (b) Effect of side-interpolater. | |
| --- | --- |
| Side-interpolater | FGD↓ |
| × | 54.08 |
| ✓ | **46.42** |

| (c) Effect of masked shortcut. | |
| --- | --- |
| Masked shortcut | FGD↓ |
| × | 49.77 |
| ✓ | **46.42** |

| (d) Effect of Full/Unmasked Features. | |
| --- | --- |
| Imput type | FGD↓ |
| Full+Unmasked | **46.42** |
| Full | 55.08 |
| Unmasked | 96.63 |

| (e) Effect of wider ratio. | |
| --- | --- |
| Wider ratio | FGD↓ |
| ✓ | **46.42** |
| × | 47.75 |

| (f) Effect of blocks in side-interpolator. | |
| --- | --- |
| Number | FGD↓ |
| 1 | **46.42** |
| 2 | 46.85 |
| 3 | 55.28 |

**Effectiveness of multi-modal fusion and mask modeling.** We conduct a quantitative comparison with and without multi-modal fusion module and mask modeling. Table 3 indicates that mask modeling reduce the Diversity score, this may be attributed to the enhanced consistency among different gestures. However, the realism of the generated gestures is significantly improved.

**Effectiveness of Different Feature Processing.** To evaluate the impact of our proposed feature processing technique, we establish a baseline by excluding the feature processing component from our MDT-A2G. We subsequently incorporate different feature processing methods for comparison: AdaLN-Zero is the method utilized by DiT [29]. Table 6 demonstrates that our proposed multi-modal fusion module shows the best performance. Notably, the use of AdaLN-Zero can not enhance results, potentially due to the introduction of excessive conditional information. Detailed network architecture diagrams for these methods are included in the appendix.

**Effectiveness of different wider mask ratio.** We use different wider mask ratios [10], where "wider" refers to masking the features within the range $[\rho, \rho + 0.2]$. Table 5 provides a comparison of different wider mask ratios. Notably, MDT-A2G-B achieves optimal performance with a mask ratio of 40%. A higher mask ratio might limit the network to reconstructing masked features, reducing diversity. Our experiments show that using varied mask ratios, instead of a fixed one, improves model robustness and adaptability across different scenarios. As shown in Table 7e, removing a wider range of mask ratios led to an increase in FGD to 47.75. Implementing variable mask ratios helps the network learn to reconstruct and generate features under varied levels of information availability, thereby improving its ability to handle real-world data.

**Effectiveness of decoder depth.** As shown in Table 7a, optimal results are obtained when the number of transformer blocks of decoder is set to 4. This suggests that using a deeper decoder enhances the ability to assimilate and process correlated information across different features, thereby improving overall model effectiveness.

**Effectiveness of Side-Interpolator.** Table 7b illustrates the impact of the side-interpolator on FGD Score. Side-Interpolator helps maintain uniformity in feature processing, which is critical for the accurate reconstruction of the input data. Additionally, as depicted in Table 7f, we investigate the effect of different number of side-interpolators. Optimal results are obtained with a single side-interpolator. Increasing side-interpolators can homogenize semantic information between learnable masks and unmasked tokens, potentially reducing effective reconstruction of masked tokens, impairing feature differentiation, and degrading performance.

**Effectiveness of masked shortcut.** Table 7c shows our investigation into how masked shortcuts impact network performance. Incorporating masked shortcuts ensures that the learnable mask parameters in the side-interpolator do not interfere with unmasked tokens, facilitating more effective information flow. This design prevents the dilution of important features during encoding, enhancing the model's overall integrity and effectiveness.

**Effectiveness of Full/Unmasked Features.** As shown in Table 7d, we analyzed the scenario where only unmasked features are fed into the MDT, rather than both unmasked features and all features. This practice can inadvertently cause the network to focus on predicting the masked features, potentially harming performance. Balancing the masking process is crucial to optimize the network's ability to learn from both masked and unmasked features.

## 6 Conclusion

We introduce the novel MDT-A2G, a masked diffusion transformer for co-speech gesture generation. Benefited by the unique mask modeling scheme, MDT-A2G enhances stronger temporal and semantic relation learning among gestures, thereby accelerating the learning process. The integration of multi-modal information has proven effective in generating more realistic gestures. Besides, our efficient inference strategy has reduced denoising computation, resulting in a 5.7× speedup with minimal performance degradation. Experimental results confirm MDT-A2G's superiority in gesture generation, with significantly faster training and inference speeds than traditional counterparts. This research has bridged a significant gap in co-speech gesture generation, opening avenues for future studies.

## 7 LIMITATIONS

DiT showcases strong scalability in image generation, and we aim to explore similar scalability in the A2G domain. However, we haven't observed the benefits of MDT-A2G's scalability yet. One possible reason is insufficient training data; as network complexity grows, so should the dataset. These are hypotheses that require further validation.

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
