# OpenReview forum: "MDT-A2G: Exploring Masked Diffusion Transformers for Co-Speech Gesture Generation"
_acmmm.org/ACMMM/2024/Conference — MM2024 Poster_

### Official Review · Reviewer_W5R9 · 2024-04-29

**Rating:** 2
**Confidence:** 3

**Summary:**

The paper is about the exploration of a novel Masked Diffusion Transformer (MDT-A2G) for co-speech gesture generation. It aims to improve the generation of high-quality human gestures that are synchronized with speech audio by leveraging the capabilities of the Transformer architecture. The MDT-A2G model incorporates a mask modeling scheme to enhance the learning of temporal relations among gesture sequences, leading to more coherent and realistic motions.
Additionally, the paper proposes an efficient inference strategy that speeds up the denoising process by using previously calculated results, achieving faster training and inference speeds compared to traditional diffusion transformers.

**Strengths:**

The paper presents a complete set of experiments and conducts ablation studies in an organized manner to analyze the outcomes resulting from the model enhancements. The final results demonstrated that the improved model outperformed other variants in the ablation studies.

**Limitations:**

I do not think that the use of a masked diffusion model here is a valid innovation point. According to the paper, in fact, any modeling between modalities that requires semantic association could potentially be improved by adding masks to force the model to pay more attention to semantic information. Alternatively, is this paper the first work in the field to add masks?
It appears as though you have first discovered an approach that could potentially be incorporated and then put in the effort to demonstrate its efficacy. This is instead of initially proving through experiments that there is a particular problem within the domain, and subsequently proposing a solution targeted at this issue. To put it another way, if the problem you are addressing is to enhance the speed of generation and the realism of gesture generation, why is it necessarily MDT? Why not some other approach?

**Suitability:**

2

---

### Official Review · Reviewer_C2hv · 2024-05-12

**Rating:** 4
**Confidence:** 2

**Summary:**

They propose MDT-A2G, a masked diffusion transformer for co-speech gesture generation. the masked diffusion model takes multi-modal conditions and partially masked noisy features as input. Their experiments demonstrate that MDT-A2G can generate high-quality, diverse
and temporally coherent human gestures with high efficiency.

**Strengths:**

- Well-designed architecture
- Effective training framework, masked diffusion model. This can be applied to other modalities
- Comprehensive evaluations, including ablation studies
- Well-written manuscript

**Limitations:**

- Qualitative examples are great, but I wonder why some models introduced in Section 2 are not compared in the experiments. I'm not familiar with speech-to-gesture generation. I'd appreciate it if the authors would explain it. I will raise my rating if I am convinced.
- By making the code publicly available, the authors can contribute more to the community.

**Suitability:**

3

---

### Official Review · Reviewer_7QWG · 2024-05-18

**Rating:** 3
**Confidence:** 3

**Summary:**

The paper attempts the task of co-speech gesture generation. For this task, the authors propose leveraging the masked diffusion transformer as the backbone to improve the generation diversity and devise a sampling strategy to speedup the generation process. Specifically, the proposed MDT-A2G first fuse the timesteps, emotion, speaker ID, text transcript, audio and noisy gesture as the diffusion latents. Then masked diffusion transformer is adopted for denoising prediction. Finally, to overcome the inefficiency of diffusion models, the authors propose a new skip-sampling method to accelerate the generation.

**Strengths:**

1. The paper is easy to understand.
2. The proposed MDT-A2G improves the performance on the task of co-speech gesture generation.
3. The authors have conduct comprehensive experiments, especially for ablation studies to evaluate the effectiveness of multiple modules.

**Limitations:**

1. The paper has limited novelty. One of the main contributions is to utilise masked diffusion transformer in the task co-speech gesture generation. However, I haven't seen any modification or improvement compared to original MDT structure. Therefore, the training acceleration compared to DSG+ also benefits the utilisation of MDT. I am not meaning that directly utilising MDT gives sufficient reason to get rejected. Demonstrating the generalization of MDT in different domains also has the merit. But from the results in Table 1, as MDT-A2G cannot consistently exceed previous approaches, some moderate modifications on MDT or the pipeline may be necessary.
2. Several important details are not well explained. From my interpretation, the conditions like audio, text are sequential data while speaker ID, emotion are scalars. The explanations on how to concatenate these data together are absent. Moreover, the audio and text may have different lengths. How to align these two modalities? Moreover, do the audio condition and output have the same lengths / frames? There are no notations regarding the frames of inputs/ latents / outputs. It is generally difficult to generate long-term sequences. However, the details on training duration / generation duration are also absent. Additionally, style loss and emotion loss are not well explained.
3. I also have some minor concerns. I understand the proposed scaling-aware accelerated sampling process may be one contribution / technical novelty in this paper. However, this strategy seems to be independent from the task of co-speech gesture generation. Why not consider other acceleration sampling strategies, such DDIM / series of DPM solvers, etc? Will the proposed method have advantages over these solvers? In Table 2, when using less inference steps (acceleration ratio is equal to 1:20 / 1:25), why the performance in terms of SRGR and BeatAlign retain the similar levels? In Table 7, why the performance is not improved with the model size scales to larger scale?

**Suitability:**

3

---

### Official Review · Reviewer_F8yB · 2024-05-21

**Rating:** 5
**Confidence:** 4

**Summary:**

"MDT-A2G: Exploring Masked Diffusion Transformers for Co-Speech Gesture Generation" introduces a novel approach using Masked Diffusion Transformers (MDT) for generating human gestures that align with speech. The proposed model, MDT-A2G, enhances the contextual reasoning of temporally aligned speech-driven gestures by employing a mask modeling scheme, which strengthens temporal relation learning among gesture sequences. This model integrates multi-modal information, including audio, text, emotion, and identity, and proposes an efficient inference strategy to speed up the denoising process.

* Novel Framework: Introduction of MDT-A2G, a framework leveraging masked modeling combined with diffusion transformers for co-speech gesture generation.
* Multi-modal Integration: Incorporation of text, emotion, and identity information to improve the contextual understanding of gestures.
* Efficient Inference Strategy: A sampling acceleration technique that reduces inference time while maintaining performance.
* Improved Performance: Demonstrated superior performance in generating realistic and diverse gestures with faster training and inference speeds compared to existing methods.

**Strengths:**

* Innovative Use of Masked Modeling: The application of masked modeling to diffusion transformers for gesture generation is novel and enhances temporal relation learning.
* Multi-modal Fusion: Effective integration of various modalities (audio, text, emotion, identity) results in more realistic and contextually appropriate gestures.
* Efficiency: The proposed sampling acceleration method significantly reduces inference time, making the model more practical for real-time applications.
* Comprehensive Evaluation: Extensive experiments and ablation studies demonstrate the model's superior performance in terms of gesture quality, diversity, and training efficiency.

**Limitations:**

* Complexity of Model: The model's complexity, with multiple components like SEAFusion and various feature extractors, might pose challenges for implementation and require significant computational resources.
* Evaluation Metrics: While the paper uses metrics like FGD, Diversity, SRGR, and BeatAlign, a more detailed analysis of how these metrics correlate with human perceptual quality could strengthen the evaluation.
* Generalization: The model is trained and tested on the BEAT dataset, which, although comprehensive, may not cover all variations in human gestures and speech patterns. Testing on additional datasets could further validate the model's robustness.
* Ablation Studies: Although thorough, the ablation studies could include more detailed analysis on the trade-offs between model complexity and performance improvements, particularly in practical deployment scenarios.
* Subjective Evaluation: Based on the findings from the GENEA Challenge, the reliance on objective metrics alone may not fully capture the perceptual quality of the generated gestures. The supplementary video suggests that the performance of MDT-A2G is not significantly different from baseline models, highlighting the need for subjective evaluation metrics and corresponding experiments to assess human perception.
* Comparative Analysis: There should be more extensive comparisons with other baseline models. Including a wider range of baseline models in the comparative analysis would provide a clearer understanding of the advantages and limitations of MDT-A2G.

**Suitability:**

3

---

### Meta-Review · Area_Chair_idZH · 2024-07-04

**Recommendation:** Accept (Poster)
**Confidence:** 5

**Metareview:**

The manuscript introduces MDT-A2G, a novel Masked Diffusion Transformer for co-speech gesture generation, which integrates multi-modal information including text, emotion, and identity. This model enhances contextual reasoning of temporally aligned speech-driven gestures through a mask modeling scheme, resulting in coherent and realistic motions. The proposed approach boasts significant improvements in learning and inference speeds compared to traditional diffusion transformers. The manuscript should incorporate detailed responses to enhance clarity and practicality, addressing the points raised by the reviewers to improve its final version.